# 1 Rapidly Changing Lake-Terminating Glaciers in High

## 2 Mountain Asia: A Dataset from 1990 to 2022

- Yunyi Luo<sup>1,2</sup>, Qiao Liu<sup>1</sup>, Xueyuan Lu<sup>1,2</sup>, Yongsheng Yin<sup>1,2</sup>, Jiawei Yang<sup>1,2</sup>, Xuyang
- Lu
- 1 Institute of Mountain Hazards and Environment, Chinese Academy of Sciences, Chengdu 610041,
- China
- 2 College of Resources and Environment, University of Chinese Academy of Sciences, Beijing 100049,
- China
- Correspondence: Qiao Liu; liuqiao@imde.ac.cn
- Abstract. Lake-terminating glaciers (LTGs) typically exhibit higher rates of retreat and thinning 11 compared to land-terminating glaciers. However, a comprehensive inventory for LTGs and their 12 associate proglacial lakes across High Mountain Asia (HMA) is currently lacking, limiting further 13 understanding of their spatial heterogeneity in glacier change. This study employs a semi-automated 14 identification method, coupled with rigorous visual inspection, to construct a comprehensive inventory 15 of LTGs and proglacial lakes in HMA for 1990 and 2022. Our data indicate that, by 2022, HMA hosted 16  $1740 \text{ LTGs} (5082.08 \pm 13.15 \text{ km}^2)$ , among which  $667 \text{ glaciers} (3454.59 \pm 12.43 \text{ km}^2)$  remained in contact 17 with proglacial lakes since 1990, 1073 (1627.49  $\pm$  4.30 km<sup>2</sup>) are newly developed and 468 (960.13  $\pm$ 18 3.18 km²) had disconnected from proglacial lakes during the investigation period. Accordingly, 645 19 proglacial lakes (207.18  $\pm$  0.82 km<sup>2</sup>) remained in contact with ice, 1123 new lakes (54.85  $\pm$  0.35 km<sup>2</sup>) 20 formed, and 485 lakes (45.31 ± 0.34 km<sup>2</sup>) detached from ice (including 25 disappeared). During the past 21 32 year, the total area of proglacial lakes increased by 138.19 ± 1.18 km² (81.7%), alongside a glacier 22 area loss of  $324.43 \pm 19.23 \text{ km}^2$  (5.1%). The southern regions of HMA, particularly the Hindu Kush, 23 Himalayas, Nyainqentanglha, and Gangdise Mountains, exhibiting the highest concentration and rapidest 24 changes of the glacier-lake system. We hope that this dataset will improve our understanding of mountain 25 glacier-lake interactions, water availability, as well as glacier-related hazards in HMA. 26 The dataset is available at https://doi.org/10.5281/zenodo.17369580 (Luo and Liu, 2025).

## 1 Introduction

al., 2020; Truffer and Motyka, 2016; Chernos et al., 2016) and are a primary driver of spatial heterogeneity in glacier responses to climate change (Brun et al., 2019; Maurer et al., 2019). Proglacial lakes typically form behind end or lateral moraines, on debris-covered glaciers often developed through the coalescence of multiple supraglacial ponds near the glacier terminus (Carrivick and Tweed, 2013; Quincey et al., 2007; Mertes et al., 2017). The influence of lake water on glacier change operates primarily through two mechanisms: (1) thermal undercutting by lake water (Truffer and Motyka, 2016) and calving at the glacier front (Benn et al., 2007a), which together accelerate subaquatic and frontal ablation; and (2) when glacier termini come into contact with sufficiently deep water, the buoyancy of the lake reduces basal effective pressure, thereby enhancing glacier flow and dynamic thinning (Sugiyama et al., 2011; Sutherland et al., 2020; Benn et al., 2007b). (Sato et al., 2022; Tsutaki et al., 2019; Tsutaki et al., 2017). Observations indicate that LTGs in HMA have mass loss rates 18-97% higher than the regional average (Brun et al., 2019), and under comparable geographic conditions, their flow velocities are typically two- to threefold greater than those land-terminating counterparts. Furthermore, Zhang et al. (2023) reported that existing geodetic methods, by failing to account for the replacement of glacier ice by lake water, underestimate the mass loss of Himalayan LTGs by approximately 6.5%. HMA encompassing the entire Tibetan Plateau and its surroundings contains the largest concentration of mid-latitude mountain glaciers on Earth. Driven by ongoing global warming, glaciers in HMA have undergone a persistent negative mass balance, with an average mass loss rate of -20.1 Gt a-1 during 2000-2019 (Hugonnet et al., 2021). Glacier meltwater has driven substantial runoff and facilitated the formation and expansion of glacial lakes. From 1990 to 2018, the number of glacial lakes in HMA increased by 11%, and their total area expanded by 15% (Wang et al., 2020). The ongoing increase in both the number and extent of proglacial lakes underscores the critical need for a comprehensive assessment of lake-terminating glacier-proglacial lake systems in HMA. Such an evaluation is essential for elucidating feedback between the lake and ice, forecasting their responses to future climate change, and informing evidence-based strategies for water resource management and disaster risk mitigation. Although several regional-scale glacial lake inventories have been published in

Proglacial lakes in direct contact with glacier termini play a critical role in glacier evolution (Liu et

recent years (Wang et al., 2020; Chen et al., 2021; Zhang et al., 2015; Worni et al., 2013; Salerno et al., 2012; Shugar et al., 2020), most datasets do not distinguish the contact status and its change between glaciers and proglacial lakes. Moreover, there is currently no comprehensive inventory of lake-terminating glacier-proglacial lake systems covering the entire HMA, and their spatiotemporal evolution remains poorly understood. Therefore, this study aims to construct a dataset of LTGs and proglacial lakes for HMA based on multi-source remote sensing data, thereby filling this research gap and providing fundamental database to support studies on regional glacier change, water resource assessment, disaster management, and glacier hydrology.

## 2 Study area

Figure 1: Location of HMA and distribution of LTGs. Glacier outlines from the Randolph Glacier Inventory (RGI v7.0). Types of LTGs are shown in Table 1.

High Mountain Asia (HMA), encompassing the Tibetan Plateau and its surrounding ranges-including the Himalayas, Karakoram, and Pamir Plateau, etc.-constitutes the most glacier-rich region in the mid-latitudes (Figure 1). HMA lies between 26°-45°N and 67°-105°E. It has an average elevation of approximately 4,500 m. The region features a complex topography. This topography is characterized by

higher elevations in the northwest and lower elevations in the southeast. It comprises a network of interwoven mountain ranges, valleys, and river systems. The dominant orographic orientation is east-west. The Tanggula Shan, located in the central part of the region, rise above 6,000 m, while the Himalayas contain 15 peaks exceeding 8000 m, and most peaks on the northern plateau surpass 6500 m. North-south trending ranges are mainly distributed in the southeastern Tibetan Plateau and the Hengduan Shan, forming the geomorphological framework of the region and controlling the overall topographic configuration of the plateau.

Climatically, the southern part of HMA is dominated by the South Asian and East Asian monsoons, bringing abundant precipitation, whereas the northern and western sectors are under the influence of the mid-latitude westerlies, characterized by arid conditions and scarce precipitation (Yao et al., 2012). This pronounced north-south climatic contrast results in a highly heterogeneous spatial pattern of glacier accumulation and ablation across the region. HMA serves as the source region for several major Asian rivers, including the Yellow River, Yangtze River, Yarlung Tsangpo, Indus, Ganges, Salween, Mekong, and Irrawaddy, which are vital for downstream hydrological processes and water resource availability. According to the Randolph Glacier Inventory (RGI 7.0), HMA hosts 94,058 modern glaciers, covering approximately 99,468.4km², making it the most extensively glacierized region outside polar areas. Most glaciers in HMA are undergoing retreat (Brun et al., 2017; Hugonnet et al., 2021). However, slight mass gains have been observed in parts of the Karakoram and western Kunlun ranges (Gardelle et al., 2012; Kääb et al., 2015), though recent studies suggest this trend may be diminishing (Hugonnet et al., 2021).

## 3 Data and methodology

## 3.1 Extraction of proglacial lake outlines

Before developing a comprehensive inventory of LTGs shown in Figure 1, we first generated a proglacial lake dataset using an automated delineation workflow within the Google Earth Engine (GEE) platform. We used Landsat imagery from the Thematic Mapper (TM) and Operational Land Imager (OLI) sensors, selected for their long-term record (since 1972), 30 m resolution, global coverage, and open access. All images were pre-processed in GEE, including radiometric, atmospheric, and geometric corrections. To minimize seasonal variability and the presence of snow and ice, we selected images

| acquired from July to November. Two-time windows were defined: $1990\pm 2$ years (historical) and                                                                                                                                                                                                                                                                                                                                                                                                                                                                                                                                                                                                                                                                                                                                                                                                                                                                                                                                                                                                                                                                                                                                                                                                                                                                                                                                                                                                                                                                                                                                                                                                                                                                                                                                                                                                                                                                                                                                                                                                                              |
|--------------------------------------------------------------------------------------------------------------------------------------------------------------------------------------------------------------------------------------------------------------------------------------------------------------------------------------------------------------------------------------------------------------------------------------------------------------------------------------------------------------------------------------------------------------------------------------------------------------------------------------------------------------------------------------------------------------------------------------------------------------------------------------------------------------------------------------------------------------------------------------------------------------------------------------------------------------------------------------------------------------------------------------------------------------------------------------------------------------------------------------------------------------------------------------------------------------------------------------------------------------------------------------------------------------------------------------------------------------------------------------------------------------------------------------------------------------------------------------------------------------------------------------------------------------------------------------------------------------------------------------------------------------------------------------------------------------------------------------------------------------------------------------------------------------------------------------------------------------------------------------------------------------------------------------------------------------------------------------------------------------------------------------------------------------------------------------------------------------------------------|
| $2022 \pm 1$ year (recent). Due to limited image availability around 1990, imagery from 1993 to 1996 was                                                                                                                                                                                                                                                                                                                                                                                                                                                                                                                                                                                                                                                                                                                                                                                                                                                                                                                                                                                                                                                                                                                                                                                                                                                                                                                                                                                                                                                                                                                                                                                                                                                                                                                                                                                                                                                                                                                                                                                                                       |
| used to supplement data gaps. A 2 km buffer around each glacier was applied to focus on potential ice-                                                                                                                                                                                                                                                                                                                                                                                                                                                                                                                                                                                                                                                                                                                                                                                                                                                                                                                                                                                                                                                                                                                                                                                                                                                                                                                                                                                                                                                                                                                                                                                                                                                                                                                                                                                                                                                                                                                                                                                                                         |
| contact proglacial lakes. Cloud contamination was reduced using the CFMask algorithm (Foga et al.,                                                                                                                                                                                                                                                                                                                                                                                                                                                                                                                                                                                                                                                                                                                                                                                                                                                                                                                                                                                                                                                                                                                                                                                                                                                                                                                                                                                                                                                                                                                                                                                                                                                                                                                                                                                                                                                                                                                                                                                                                             |
| 2017) to detect and mask clouds and shadows, followed by compositing cloud-free mosaics (Figure 2ab).                                                                                                                                                                                                                                                                                                                                                                                                                                                                                                                                                                                                                                                                                                                                                                                                                                                                                                                                                                                                                                                                                                                                                                                                                                                                                                                                                                                                                                                                                                                                                                                                                                                                                                                                                                                                                                                                                                                                                                                                                          |
| In total, 4570 Landsat TM scenes were used for the 1990 period and 5493 OLI scenes for the 2020 period                                                                                                                                                                                                                                                                                                                                                                                                                                                                                                                                                                                                                                                                                                                                                                                                                                                                                                                                                                                                                                                                                                                                                                                                                                                                                                                                                                                                                                                                                                                                                                                                                                                                                                                                                                                                                                                                                                                                                                                                                         |
| (Figure 2cd).                                                                                                                                                                                                                                                                                                                                                                                                                                                                                                                                                                                                                                                                                                                                                                                                                                                                                                                                                                                                                                                                                                                                                                                                                                                                                                                                                                                                                                                                                                                                                                                                                                                                                                                                                                                                                                                                                                                                                                                                                                                                                                                  |
| Glacial lake extents were delineated using an automated mapping algorithm based on hierarchical                                                                                                                                                                                                                                                                                                                                                                                                                                                                                                                                                                                                                                                                                                                                                                                                                                                                                                                                                                                                                                                                                                                                                                                                                                                                                                                                                                                                                                                                                                                                                                                                                                                                                                                                                                                                                                                                                                                                                                                                                                |
| image segmentation and terrain analysis (Li and Sheng, 2012; Zhang et al., 2017). To reduce the influence                                                                                                                                                                                                                                                                                                                                                                                                                                                                                                                                                                                                                                                                                                                                                                                                                                                                                                                                                                                                                                                                                                                                                                                                                                                                                                                                                                                                                                                                                                                                                                                                                                                                                                                                                                                                                                                                                                                                                                                                                      |
| of mountain shadows, pixels with slopes $>20^\circ$ or shaded relief values $

Figure 2: The number of usable pixels remaining in the study area after cloud removal during 1988–1996 (a) and 2021–2023 (b). Temporal distribution of the number of images used, by year (c) and by month (d).

## 3.2 Mapping of LTGs

114 115

116

117

118

119

120

121

In this study, LTGs are defined as glaciers that develop proglacial lakes along the direction of ice flow and are in direct contact with these lakes. The proglacial lake dataset was cross-referenced with the RGI 7.0 glacier inventory to identify LTGs. Results were refined through detailed visual inspection and manual correction using multi-source data, including Landsat and Planet Labs imagery, online maps (e.g.,

https://doi.org/10.5194/essd-2025-596 Preprint. Discussion started: 20 October 2025 © Author(s) 2025. CC BY 4.0 License.

122

123

124

125

126127

128

129

130

131

Google Earth, Esri basemap), and existing glacial lake datasets (Wang et al. 2020, Chen et al. 2021, Zhang et al. 2023). The identification of glacier-lake contact followed a two-step procedure. (1) Preliminary screening: A 500 m buffer was applied to assess spatial intersections between glacier boundaries and proglacial lakes, identifying potentially connected glacier-lake pairs. (2) Manual verification: Different criteria were applied for different periods. For the year 2020, multi-source moderate-to-high resolution imagery (e.g., Planet Labs, Landsat, Google Earth, Esri basemaps) was used. Glacier-lake contact was confirmed when proglacial lakes overlapped with glacier terminus and exhibited diagnostic geomorphic features, such as terminal ice cliffs or transverse crevasses perpendicular to the flow direction. Due to limited data availability and the relatively coarse spatial resolution of Landsat imagery (30 m) in 1990, direct identification of LTGs for that year involved considerable uncertainty, particularly for small glaciers, where boundary errors increase with decreasing glacier area. To address this, a temporal cross-validation approach was employed. Glaciers with ambiguous contact in 1990 were classified as interacting if satellite imagery from 1990 to 2022 showed lake expansion toward the glacier terminus. Based on the temporal evolution of glacier-lake contact, LTGs were categorized into three types (Table 1): (1) terminus persistent contacting with proglacial lake (Type 1); (2) terminus experencing transition from supraglacial lake to proglacial lake (Type 2); and (3) terminus detaching from proglacial lake (Type 3).

# Table 1 The classification system of glaciers is based on the dynamic changes in glacier-lake contact. The basemap is derived from Landsat imagery.

| Types  | Characteristics                                                                                                      |               |            |           |
|--------|----------------------------------------------------------------------------------------------------------------------|---------------|------------|-----------|
| Type 1 | Persistent contact<br>between glacier and<br>lake from 1990 to<br>2022.<br>Case location:<br>94,51053E,<br>30,63100N | 0 2 km        | 0 200      | 16-Antown |
| Type 2 | Transition from supraglacial lake to proglacial lake from 1990 to 2022.  Case location: 88.23816E, 27.81772N         | 11. day. 920) | 0 1 m      | 0 5 mm    |
| Type 3 | Detachment of the proglacial lake from the parent glacier from 1990 to 2022.  Case location: 85.84583E, 28.20793N    | (R)200 A      | POCHED THE | 640999    |

## 3.3 Uncertainty estimates

When interpreting glacial lake and glacier boundaries using remote sensing data, errors are inevitable even when manual visual delineation is applied. These errors are typically associated with various factors related to image quality, such as spatial resolution, cloud cover, mountain shadows, and subjective interpretation biases. Previous studies have reported that the area error in delineating glacier or glacial lake boundaries from remote sensing imagery is approximately  $\pm 0.5$  pixels, depending on the quality of the imagery. The uncertainty ( $\delta$ ) and relative error ( $E_l$ ) of glacial lake area was estimated using the equation (Hanshaw and Bookhagen, 2014):

$$\delta = \frac{P}{G} \times \frac{G^2}{2} \times 0.6872 \tag{3}$$

$$E_l = \frac{\delta}{A} \times 100\% \tag{4}$$

where P is the perimeter of the glacial lake, and A is the glacial lake area.

The uncertainty ( $\lambda$ ) and relative error ( $E_g$ ) in glacier area was estimated to using the equation (Bolch et al., 2010):

$$\lambda = N \times \frac{G^2}{2} \tag{1}$$

 $E_g = \frac{\lambda}{S} \times 100\% \tag{2}$ 

where N is the total count of pixels along the outline of ice coverage, G is the spatial resolution of the images used, and S is the glacier area.

## 3.4 Attributes of inventory data

In this inventory, 9 attribute fields (Table 2) were recorded for the LTG, including a unique identifier, type, associated mountain range, area, mapping uncertainty, location (longitude and latitude), RGI7 ID, and feature code. Similarly, the proglacial lake inventory contains 9 attribute fields (Table 3), including a unique identifier, associated mountain range, type, mapping uncertainty, location (longitude and latitude), feature code, and a flag indicating whether the lake has disappeared. Both LTG and proglacial lake datasets include data for two time periods: 1990 and 2022, with identical attributes for both periods. The unique identifier is an automatically generated sequential integer, while the feature code follows the formats GmmmmmEnnnnnN (Feature\_ID) for glaciers and GLmmmmmmEnnnnnN (Feature\_ID) for lakes, where G denotes glacier, GL denotes glacier lake, m and n represent the longitude and latitude multiplied by 1000, respectively, and E and N indicate east longitude and north latitude. Identical LTGs and proglacial lakes share the same feature code (Feature\_ID) to facilitate data linkage. Area and perimeter are calculated automatically from the feature geometry. The type of classification follows the criteria described in Section 3.2. Each feature's associated mountain range is determined by overlaying with mountain range boundaries, and mapping uncertainty is estimated according to Section 3.3.

## Table 2 Attributes of the glacier dataset

| Filed name  | Type      | Description                                                                                                          |
|-------------|-----------|----------------------------------------------------------------------------------------------------------------------|
| UID         | Object ID | Unique code (Number)                                                                                                 |
| Type        | String    | The classification of glaciers based on the relationship of interaction between glaciers and glacial lakes (Table 1) |
| Mountain    | String    | Mountain name where the glaciers is in                                                                               |
| Area        | Double    | Area of glacier coverage(km²)                                                                                        |
| Error       | Double    | Area uncertainty of glacier mapping(km²)                                                                             |
| Latitude    | String    | Latitude of the centroid of glacier                                                                                  |
| Longitude   | String    | Longitude of the centroid of glacier                                                                                 |
| rgi_id      | String    | RGI 7.0 id                                                                                                           |
| Feature _ID | String    | GmmmmmEnnnnN                                                                                                         |

186 187

## Table 3 Attributes of the proglacial lake dataset

| Filed name  | Type      | Description                                                                                                               |
|-------------|-----------|---------------------------------------------------------------------------------------------------------------------------|
| UID         | Object ID | Unique code (Number)                                                                                                      |
| Туре        | String    | The classification of glacial lakes based on the relationship of interaction between glaciers and glacial lakes (Table 1) |
| Mountain    | String    | Mountain name where the glacial lake is in                                                                                |
| Area        | Double    | Area of glacial lake coverage (km²)                                                                                       |
| Error       | Double    | Area uncertainty of glacial lake mapping (km²)                                                                            |
| Latitude    | String    | Latitude of the centroid of glacier                                                                                       |
| Longitude   | String    | Longitude of the centroid of glacier                                                                                      |
| Disappear   | String    | Whether the proglacial lake disappeared in 2022 (Y)                                                                       |
| Feature _ID | String    | GLmmmmmEnnnnnN                                                                                                            |

4 Results

## 4.1 Spatial distribution of LTGs and proglacial lakes

Based on the changes in glacier-proglacial lake contact relationships from 1990 to 2022, glaciers were classified into three types (Table 1). Among them, Type 1 and Type 2 glaciers remained in contact with proglacial lakes from 1990 to 2022 and are therefore defined as LTGs. In contrast, Type 3 glaciers had become disconnected from proglacial lakes by 2022. Accordingly, only Type 1 and Type 2 glaciers were included when analyzing the distribution and extent of LTGs in 2022. In 2022, a total of 1740 LTGs were identified, with a combined area of 5082.08 ± 13.15 km<sup>2</sup>. Concurrently, 1768 proglacial lakes were detected, with a total area of  $262.10 \pm 0.89$  km<sup>2</sup>. The discrepancy between glacier and lake counts stems from multi-lake associations per glacier and multi-glacier lakes were associated with two glaciers, and two lakes were in contact with three glaciers. The spatial distribution of LTGs in HMA shows marked heterogeneity (Figure 3). Predominantly concentrated along the southern margin, including the Himalayas, Nyainqentanglha, Gangdise Mountains, and Hindu Kush, these glaciers total 994, representing 57.13% of the study population (Figure 3b, Table A 1). The Central Himalaya hosts the highest number, with 232 glaciers (Table A 1), while the Nyainqentanglha accounts for the largest total glacier area  $(1,001.05 \pm 3.32 \text{ km}^2,\text{Figure 3c})$ . Glaciers were classified into nine size categories, ranging from  $<0.05~km^2$  to  $>100~km^2$  (Table A2). Among these, 1,095 glaciers (62.93%) are smaller than 1 km², covering  $399.05 \pm 1.07$  km<sup>2</sup> (7.85% of the total area), while 93 glaciers (5.35%) exceed 10 km<sup>2</sup>, covering 2964.68 ± 4.85 km² (58.34%). Only three glaciers exceed 100 km², spanning 785.42 ± 10.96 km². LTGs in HMA span elevations from 2,735 to 8,016 m, with a mean elevation of 5074 m (Figure 4). They are primarily concentrated between 5,000 and 6,000 m, where their combined area reaches  $3030.2 \pm 5.72$ km<sup>2</sup> (59.52% of the total glacier area). Regional variations in elevation distribution are evident (Figure 4). In the Central Himalaya, Eastern Himalaya, Gangdise Mountains, Tibetan Interior Mountains, and Western Kunlun Shan, glacier area peaks occur around 6000 meters. Proglacial lakes in HMA are predominantly concentrated along the southern margin, with 1010 lakes (57.09%) in the Himalayas, Nyainqentanglha, Gangdise Mountains, and Hindu Kush (Table A 3). The Central Himalayas host the most lakes (240), with the largest total area (86.91  $\pm$  0.54 km<sup>2</sup>, Figure 3 e). Proglacial lakes were grouped into five size categories (<0.05 to >1 km², Table A 4). Lakes smaller

| than 0.1 km² are the most abundant, totaling 1384 (78.28%) and covering a combined area of 47.12 $\pm$                                                                                                                                                                                                                                                                                                                                                                                                                                                                                                                                                                                                                                                                                                                                                                                                                                                                                                                                                                                                                                                                                                                                                                                                                                                                                                                                                                                                                                                                                                                                                                                                                                                                                                                                                                                                                                                                                                                                                                                                                         |
|--------------------------------------------------------------------------------------------------------------------------------------------------------------------------------------------------------------------------------------------------------------------------------------------------------------------------------------------------------------------------------------------------------------------------------------------------------------------------------------------------------------------------------------------------------------------------------------------------------------------------------------------------------------------------------------------------------------------------------------------------------------------------------------------------------------------------------------------------------------------------------------------------------------------------------------------------------------------------------------------------------------------------------------------------------------------------------------------------------------------------------------------------------------------------------------------------------------------------------------------------------------------------------------------------------------------------------------------------------------------------------------------------------------------------------------------------------------------------------------------------------------------------------------------------------------------------------------------------------------------------------------------------------------------------------------------------------------------------------------------------------------------------------------------------------------------------------------------------------------------------------------------------------------------------------------------------------------------------------------------------------------------------------------------------------------------------------------------------------------------------------|
| $0.30\ km^2$ . Proglacial lakes in HMA span elevations from 2684 to 6012 m, with most concentrated                                                                                                                                                                                                                                                                                                                                                                                                                                                                                                                                                                                                                                                                                                                                                                                                                                                                                                                                                                                                                                                                                                                                                                                                                                                                                                                                                                                                                                                                                                                                                                                                                                                                                                                                                                                                                                                                                                                                                                                                                             |
| between 5000 and 5700 m, where 748 lakes (42.34%) cover $106.46 \pm 0.59$ km². Regional variations in                                                                                                                                                                                                                                                                                                                                                                                                                                                                                                                                                                                                                                                                                                                                                                                                                                                                                                                                                                                                                                                                                                                                                                                                                                                                                                                                                                                                                                                                                                                                                                                                                                                                                                                                                                                                                                                                                                                                                                                                                          |
| elevation distribution are evident (Figure 5). Gangdise Mountains and Western Kunlun Shan, proglacial                                                                                                                                                                                                                                                                                                                                                                                                                                                                                                                                                                                                                                                                                                                                                                                                                                                                                                                                                                                                                                                                                                                                                                                                                                                                                                                                                                                                                                                                                                                                                                                                                                                                                                                                                                                                                                                                                                                                                                                                                          |
| $lake \ numbers \ and \ areas \ peak \ around \ 5700 \ m. \ Conversely, in the Hindu Kush, Nyainqentanglha, Tanggulanglang, Tanggulanglang, Tanggulang, Tanggulan$ |
| Shan, and Western Kunlun Shan, peak lake areas occur at lower elevations than peak lake numbers                                                                                                                                                                                                                                                                                                                                                                                                                                                                                                                                                                                                                                                                                                                                                                                                                                                                                                                                                                                                                                                                                                                                                                                                                                                                                                                                                                                                                                                                                                                                                                                                                                                                                                                                                                                                                                                                                                                                                                                                                                |
| (Figure 5).                                                                                                                                                                                                                                                                                                                                                                                                                                                                                                                                                                                                                                                                                                                                                                                                                                                                                                                                                                                                                                                                                                                                                                                                                                                                                                                                                                                                                                                                                                                                                                                                                                                                                                                                                                                                                                                                                                                                                                                                                                                                                                                    |
| Significant variations exist in the number and area distributions among glacier types in HMA. From                                                                                                                                                                                                                                                                                                                                                                                                                                                                                                                                                                                                                                                                                                                                                                                                                                                                                                                                                                                                                                                                                                                                                                                                                                                                                                                                                                                                                                                                                                                                                                                                                                                                                                                                                                                                                                                                                                                                                                                                                             |
| 1990 to 2022, Type 2 glaciers, those forming new proglacial lakes, were the most numerous (1073, Table                                                                                                                                                                                                                                                                                                                                                                                                                                                                                                                                                                                                                                                                                                                                                                                                                                                                                                                                                                                                                                                                                                                                                                                                                                                                                                                                                                                                                                                                                                                                                                                                                                                                                                                                                                                                                                                                                                                                                                                                                         |
| A 1), dominating in all regions except Altun Shan/Eastern Kunlun Shan, Qilian Shan, and Tanggula Shan.                                                                                                                                                                                                                                                                                                                                                                                                                                                                                                                                                                                                                                                                                                                                                                                                                                                                                                                                                                                                                                                                                                                                                                                                                                                                                                                                                                                                                                                                                                                                                                                                                                                                                                                                                                                                                                                                                                                                                                                                                         |
| Conversely, Type 1 glaciers have the largest total area (3454.59 $\pm$ 12.43 km²), concentrated primarily                                                                                                                                                                                                                                                                                                                                                                                                                                                                                                                                                                                                                                                                                                                                                                                                                                                                                                                                                                                                                                                                                                                                                                                                                                                                                                                                                                                                                                                                                                                                                                                                                                                                                                                                                                                                                                                                                                                                                                                                                      |
| in the Himalayas, Nyainqentanglha, Central Tien Shan, Qilian Shan, Tanggula Shan, and Western Kunlun                                                                                                                                                                                                                                                                                                                                                                                                                                                                                                                                                                                                                                                                                                                                                                                                                                                                                                                                                                                                                                                                                                                                                                                                                                                                                                                                                                                                                                                                                                                                                                                                                                                                                                                                                                                                                                                                                                                                                                                                                           |
| Shan (Table A 1). The Central Himalaya host the most glaciers across all types: 94 Type 1 (552.77 $\pm2.71$                                                                                                                                                                                                                                                                                                                                                                                                                                                                                                                                                                                                                                                                                                                                                                                                                                                                                                                                                                                                                                                                                                                                                                                                                                                                                                                                                                                                                                                                                                                                                                                                                                                                                                                                                                                                                                                                                                                                                                                                                    |
| km²), 138 Type 2 (244.80 $\pm$ 1.56 km²), and 84 Type 3 (202.67 $\pm$ 1.11 km²). All glacier types show                                                                                                                                                                                                                                                                                                                                                                                                                                                                                                                                                                                                                                                                                                                                                                                                                                                                                                                                                                                                                                                                                                                                                                                                                                                                                                                                                                                                                                                                                                                                                                                                                                                                                                                                                                                                                                                                                                                                                                                                                        |
| consistent area peaks between $5,000$ and $6,000$ m, with similar patterns across subregions (Figure 4). In                                                                                                                                                                                                                                                                                                                                                                                                                                                                                                                                                                                                                                                                                                                                                                                                                                                                                                                                                                                                                                                                                                                                                                                                                                                                                                                                                                                                                                                                                                                                                                                                                                                                                                                                                                                                                                                                                                                                                                                                                    |
| $2022, Type\ 2\ proglacial\ lakes\ were\ the\ most\ numerous\ in\ HMA\ (1123, {\color{red}{\bf Table}}\ {\color{blue}{\bf A3}}), dominating\ in\ number\ {\color{blue}{\bf B3}}$                                                                                                                                                                                                                                                                                                                                                                                                                                                                                                                                                                                                                                                                                                                                                                                                                                                                                                                                                                                                                                                                                                                                                                                                                                                                                                                                                                                                                                                                                                                                                                                                                                                                                                                                                                                                                                                                                                                                               |
| $across\ all\ regions\ except\ Altun\ Shan/Eastern\ Kunlun\ Shan,\ Qilian\ Shan,\ Karakoram,\ and\ Western\ Kunlun\ Shan,\ Gandard Shan,\ Ga$ |
| Shan. Conversely, Type 1 lakes had the largest total area (207.18 $\pm$ 0.82 km²) and accounted for the                                                                                                                                                                                                                                                                                                                                                                                                                                                                                                                                                                                                                                                                                                                                                                                                                                                                                                                                                                                                                                                                                                                                                                                                                                                                                                                                                                                                                                                                                                                                                                                                                                                                                                                                                                                                                                                                                                                                                                                                                        |
| largest share of total area in all regions except the Western Pamir, Hengduan Shan, Dzhungarsky Alatau,                                                                                                                                                                                                                                                                                                                                                                                                                                                                                                                                                                                                                                                                                                                                                                                                                                                                                                                                                                                                                                                                                                                                                                                                                                                                                                                                                                                                                                                                                                                                                                                                                                                                                                                                                                                                                                                                                                                                                                                                                        |
| and Eastern Tibetan Mountains. The central Himalaya hosted the greatest abundance of all three lake                                                                                                                                                                                                                                                                                                                                                                                                                                                                                                                                                                                                                                                                                                                                                                                                                                                                                                                                                                                                                                                                                                                                                                                                                                                                                                                                                                                                                                                                                                                                                                                                                                                                                                                                                                                                                                                                                                                                                                                                                            |
| types, with 91 Type 1 (76.89 $\pm$ 0.51 km²), 149 Type 2, and 80 Type 3 (15.70 $\pm$ 0.21 km²) lakes. The                                                                                                                                                                                                                                                                                                                                                                                                                                                                                                                                                                                                                                                                                                                                                                                                                                                                                                                                                                                                                                                                                                                                                                                                                                                                                                                                                                                                                                                                                                                                                                                                                                                                                                                                                                                                                                                                                                                                                                                                                      |
| Eastern Himalaya had the largest Type 2 lake area (10.73 $\pm$ 0.03 km², Table A3). In HMA, the elevation                                                                                                                                                                                                                                                                                                                                                                                                                                                                                                                                                                                                                                                                                                                                                                                                                                                                                                                                                                                                                                                                                                                                                                                                                                                                                                                                                                                                                                                                                                                                                                                                                                                                                                                                                                                                                                                                                                                                                                                                                      |
| distribution of proglacial lake types is generally consistent, with peak numbers between $5000$ and $5700$                                                                                                                                                                                                                                                                                                                                                                                                                                                                                                                                                                                                                                                                                                                                                                                                                                                                                                                                                                                                                                                                                                                                                                                                                                                                                                                                                                                                                                                                                                                                                                                                                                                                                                                                                                                                                                                                                                                                                                                                                     |
| m and peak areas between 4700 and 5400 m (Figure 5). However, regional variations are observed in the                                                                                                                                                                                                                                                                                                                                                                                                                                                                                                                                                                                                                                                                                                                                                                                                                                                                                                                                                                                                                                                                                                                                                                                                                                                                                                                                                                                                                                                                                                                                                                                                                                                                                                                                                                                                                                                                                                                                                                                                                          |
| elevation distribution of lake numbers for different lake types. Specifically, in the Nyainqentanglha                                                                                                                                                                                                                                                                                                                                                                                                                                                                                                                                                                                                                                                                                                                                                                                                                                                                                                                                                                                                                                                                                                                                                                                                                                                                                                                                                                                                                                                                                                                                                                                                                                                                                                                                                                                                                                                                                                                                                                                                                          |
| $region, Type\ 2\ proglacial\ lakes\ exhibit\ a\ higher\ peak\ number\ range,\ between\ 5200\ and\ 5400\ m.\ Regarding$                                                                                                                                                                                                                                                                                                                                                                                                                                                                                                                                                                                                                                                                                                                                                                                                                                                                                                                                                                                                                                                                                                                                                                                                                                                                                                                                                                                                                                                                                                                                                                                                                                                                                                                                                                                                                                                                                                                                                                                                        |
| area-elevation patterns, certain subregions display lower peak elevations, encompassing Type 2 lakes in                                                                                                                                                                                                                                                                                                                                                                                                                                                                                                                                                                                                                                                                                                                                                                                                                                                                                                                                                                                                                                                                                                                                                                                                                                                                                                                                                                                                                                                                                                                                                                                                                                                                                                                                                                                                                                                                                                                                                                                                                        |

| 245 | Kush, Nyainqentanglha, and Tanggula Shan (Figure 4).                                                            |
|-----|-----------------------------------------------------------------------------------------------------------------|
| 246 | 4.2 Temporal changes in LTGs and proglacial lakes                                                               |
| 247 | From 1990 to 2022, glacier size has been continuously shrinking (Figure 3d). The total                          |
| 248 | area of all glacier types decreased by approximately 324.43 $\pm$ 19.22 km², with Type 1 glaciers               |
| 249 | experiencing the largest absolute loss of $137.46 \pm 17.62 \; km^2$ , accounting for $42.37\%$ of the total    |
| 250 | reduction (Table A 5). The Central Himalay showed the most pronounced absolute area loss,                       |
| 251 | with a decrease of $74.46 \pm 3.46 \; km^2$ , while the Hengduan Shan exhibited the highest relative            |
| 252 | shrinkage at 16.42%. The Central Himalaya also recorded the largest absolute losses for all                     |
| 253 | three glacier types, with reductions of $37.20\pm3.91~km^2$ for Type 1, $20.13\pm2.26~km^2$ for Type            |
| 254 | 2, and 17.13 $\pm$ 1.62 $km^2$ for Type 3 glaciers. In contrast, the Hengduan Shan had the highest              |
| 255 | relative losses for all three types, at 25.34%, 13.95%, and 17.37%, respectively (Table A 5).                   |
| 256 | Small glaciers ( $<$ 0.5 km $^2$ ) exhibited a significant increase in number, particularly those               |
| 257 | smaller than 0.05 km², which grew by 51 in count with a total area increase of 1.68 $\pm$ 0.08 km               |
| 258 | $^2$ (Table A 6). In contrast, glaciers in the $0.5-50~\mathrm{km^2}$ range showed a declining trend in number. |
| 259 | Among them, glaciers sized 0.5-1 km² experienced the largest numerical decrease (-57) and                       |
| 260 | the greatest relative area loss ( $-13.56\%$ ), while those in the $1-5~\mathrm{km^2}$ range incurred the most  |
| 261 | substantial absolute area reduction, losing $97.17 \pm 3.5 \text{ km}^2$ (Table A 6).                           |
| 262 | Among the different glacier types, Type 1 glaciers experienced the greatest absolute area                       |
| 263 | loss, decreasing by 137.46 $\pm$ 17.62 km <sup>2</sup> (Table A 5). However, their relative area reduction of   |
| 264 | 3.83% was the smallest among the three types. By size class (Table A 6), Type 1 glaciers showed                 |
| 265 | the largest loss (63.39 $\pm$ 6.38 km²) in the 10–50 km² range; Type 2 glaciers experienced the                 |
| 266 | greatest reduction (52.52 $\pm$ 2.21 km²) in the 1–5 km² range. Type 3 glaciers showed the most                 |
| 267 | significant loss (27.51 $\pm$ 1.57 km²) in the 5–10 km² range. For all three types, the 0.5–1 km²               |
| 268 | size class exhibited the highest relative area reduction, at 9.06%, 15.37%, and 15.15%,                         |
| 269 | respectively.                                                                                                   |
| 270 | Between 1990 and 2022, the total area of proglacial lakes increased by $138.19 \pm 1.18 \ km^2$ ,               |
| 271 | representing a 62.09% expansion (Figure 3f and Table A2). The Central Himalaya experienced                      |

the Eastern Himalaya and Northern Tibetan Mountains, and Type 1 lakes in the Eastern Pamirs, Hindu

| 272 | the most significant absolute growth, with an increase of $42.32 \pm 0.72$ km² (70.19%), while the              |
|-----|-----------------------------------------------------------------------------------------------------------------|
| 273 | Western Pamirs recorded the fastest relative growth, surging by 210.24%. The Central                            |
| 274 | Himalaya also saw the largest area increases across all three glacier types, with growth of 30.42               |
| 275 | $\pm~0.64~km^2$ for Type 1 lakes, $10.02\pm0.16~km^2$ for Type 2, and $1.88\pm0.29~km^2$ for Type 3.            |
| 276 | Regionally, the Dzhungarsky Alatau had the highest proportional increase in Type 1 lake area                    |
| 277 | at 176.38%, whereas the Eastern Himalaya recorded the largest proportional growth for Type 3 $$                 |
| 278 | lakes at 29.48% (Table A7).                                                                                     |
| 279 | During the study period, 1123 new proglacial lakes formed, while 25 lakes disappeared.                          |
| 280 | The number of small proglacial lakes (<0.5 km²) increased significantly, especially those                       |
| 281 | smaller than 0.05 km², which increased by 702 and accounted for $64.11\%$ of the total increase                 |
| 282 | in lake numbers (Table A 8). Lakes larger than 1 km² contributed the largest increase in area                   |
| 283 | (60.44 $\pm$ 0.81 km²), accounting for 43.74% of the total area growth. Moreover, lakes smaller                 |
| 284 | than 0.05 km² had the highest proportional area growth at 114.49%. Type 1 proglacial lakes                      |
| 285 | exhibited the most significant area growth, reaching 79.36 $\pm$ 1.02 km², with a growth rate of                |
| 286 | $62.09\%$ . Among size categories, the number of Type 1 lakes increased most in the $0.05-0.1~\mathrm{km}$      |
| 287 | <sup>2</sup> range, with 49 new lakes added, while lakes larger than 1 km <sup>2</sup> showed the greatest area |
| 288 | increase at $52.07 \pm 0.79 \text{ km}^2$ and the highest proportional growth at $85.19\%$ .                    |

Figure 3: (a) Geographic extent of the mountain ranges in HMA. (b) Distribution of the three types of LTGs in 2022 and their numerical proportions across mountain regions. (c) Size distribution (Types 1 and 2) in 2022 and their area proportions. (d) Area changes of the three types of glaciers from 1990 to 2022 and their area-change proportions across mountain regions. (e) Area distribution of proglacial lakes (associated with Types 1 and 2 glaciers) in 2022 and their area proportions across mountain regions. (f) Area changes of the three types of proglacial lakes from 1990 to 2022 and their area-change proportions across mountain regions.

299

300 301

Figure 4: Area-Elevation distribution of LTGs across subregions, showing glacier area within 100 m elevation bins.

Figure 5 Distribution of proglacial lake numbers and areas across elevation ranges in each subregion. The number and area of proglacial lakes are presented within 100 m elevation bins for each subregion.

## 5 Discussion

## 5.1 Assessment of accuracy and errors

The uncertainty estimates indicate that as the glacier or lake area increases, the relative error of individual features decreases. In the study area, the total absolute area error for glaciers in 1990 and 2022 were  $\pm 13.65 \,\mathrm{km^2}$  and  $\pm 13.53 \,\mathrm{km^2}$ , respectively, with average relative errors of  $\pm 7.24\%$  and  $\pm 8.12\%$ . The relative error of glacier area shows a significant power-law relationship with the glacier size ( $y = 0.056 \times x^{-0.427}$ ,  $R^2 = 0.92$ , Figure 6a). Additionally, the total absolute area error for proglacial lakes in 1990 and 2022 were  $\pm 0.69 \,\mathrm{km^2}$  and  $\pm 0.96 \,\mathrm{km^2}$ , respectively, with average relative errors of  $\pm 21.99\%$  and  $\pm 23.69\%$ , following a similar significant power-law relationship ( $y = 0.050 \times x^{-0.463}$ ,  $R^2 = 0.94$ ,

Figure 6: Estimation of relative errors for glaciers and proglacial lakes in the study area. (a) Glaciers (b) Proglacial lakes

#### 5.2 Comparison and limitations

Publicly available data on LTGs and their proglacial lakes in HMA remain scarce, with recent datasets primarily focusing on glacial lakes. Consequently, this study selected two glacial lake datasets that partially overlap in time with our research and include proglacial lakes for comparison (Table 4). The results indicate that, within the same study area, our data closely align with those of Zhang et al. (2023). In 1990, the overlap rate of proglacial lakes between the dataset of Zhang et al. (2023) and ours exceeded 90%, while in 2020/2022, the overlap rate was 79%. In contrast, significant discrepancies were observed with the dataset of Chen et al. (2021). For the period 2017/2022, the dataset of Chen et al. (2021) identified 7850 proglacial lakes, whereas our study identified only 1,768, with an overlap rate of 67.82%. Through examining these datasets, we attribute these differences to variations in the identification of glacier-proglacial lake contact. Our study employs strict classification criteria (see Section 3.2), which

dynamics.

are reflected in three key aspects: (1) the lake must be located at the forefront of the glacier's flow direction; (2) a comprehensive evaluation of the glacier-lake contact surface based on the spatiotemporal evolution of both lake and glacier surface morphology; and (3) exclusion of ambiguous cases to ensure classification reliability. Additional factors, such as image quality, acquisition dates, and vectorization workflows, may also contribute to the observed discrepancies. A global inventory of LTGs was released in 2025 (Steiner et al., 2025). This dataset was derived from the RGI7 glacier outlines, primarily using Landsat 5-7 TM/ETM+ imagery (ca. 1998-2002), supplemented by ASTER data in some high-latitude regions. Existing regional proglacial lake inventories (when close to 2000) were also incorporated, and the identification of LTGs was conducted through manual interpretation and expert cross-validation. Based on the degree of glacier-lake contact, glaciers were classified into four types. In HMA, a total of 1912 LTGs were identified. Although the glacier termini in this dataset were delineated for  $2000 \pm 2$ , the overlap with our 2022 dataset is 47.4%. Given that our results indicate that glacier-lake contact is not always stable, differences in the timing of terminus delineation are likely the primary source of the observed discrepancies. Although this study employed standardized criteria for the qualitative identification of LTGs and their proglacial lakes, subjective factors remain challenging to eliminate entirely during remote sensing imagery analysis. Differences in how analysts interpret imagery, apply calibration standards, and process data quality directly impact the results. While measures such as independent labeling and cross-validation by multiple analysts can reduce subjective bias, uncertainties stemming from variations in individual experience, judgment criteria, and image quality remain difficult to fully resolve. Consequently, further quantification of identification criteria is of paramount importance. In the future, more refined technical approaches can optimize the identification of glacier-lake contact lines, leveraging high-resolution imagery and automated analysis tools to enhance accuracy. Additionally, quantifying the depth of glaciers within lakes will provide more precise data support. These quantitative standards not only effectively minimize human-induced variability but also significantly improve the precision of glacier-lake contact relationship assessments, laying a more reliable data foundation for subsequent studies of glacier

Table 4: Comparisons of glacial lake mapping in this study with previous studies for the similar extended region.

| Year<br>(previous/this<br>study) | Region              | Area threshod<br>(km²) | Source                  | Count (Area/km²) Previous studies | Count (Area/km²) This study | Overlap<br>count |
|----------------------------------|---------------------|------------------------|-------------------------|-----------------------------------|-----------------------------|------------------|
| 1990/1990                        | 0                   |                        | (7)                     | 651(129.76±0.89)                  | 645(122.08±0.59)            | 615(95.35%)      |
| 2020/2022                        | Greater<br>Himalaya | 0.0036                 | (Zhang et al.,<br>2023) | 1115(192.42±1.23)                 | 1029<br>(199.83±0.79)       | 841(79.11%)      |
| 2017/2022                        | НМА                 | 0.0081                 | (Chen et al.,<br>2021)  | 7850(684.62±10.06)                | 1768(262.03±0.89)           | 1199(67.82%)     |

#### **6 Conclusions**

Using Landsat imagery, we applied a semi-automated mapping approach in Google Earth Engine (GEE) to inventory proglacial lakes across High Mountain Asia (HMA) in the 1990s and 2020s, and compiled the first region-wide dataset of LTGs and their proglacial lakes. In 2022, HMA contained 1740 LTGs (5082.08  $\pm$  13.15 km<sup>2</sup>), of which 667 glaciers (3454.59  $\pm$  12.43 km<sup>2</sup>) maintained lake contact since 1990, and 1073 glaciers (1,627.49  $\pm$  4.30 km<sup>2</sup>) developed new proglacial lakes. These glaciers were mainly distributed between 2735 and 8016 m a.s.l. Additionally, 468 glaciers (960.13  $\pm$  3.18 km<sup>2</sup>) lost lake contact during the period.

A total of 1768 proglacial lakes (262.10  $\pm$  0.89 km²) were connected to glaciers in 2022, including 645 lakes (207.18  $\pm$  0.82 km²) with continuous glacier contact and 1123 newly formed lakes (54.85  $\pm$  0.35 km²). Lakes were mainly distributed between 2684 and 6012 m a.s.l. Meanwhile, 485 lakes (45.31  $\pm$  0.34 km²) lost glacier contact, with 25 disappearing entirely. From 1990 to 2022, LTGs retreated by 324.43  $\pm$  19.23 km² (–5.1%), while proglacial lake area increased by 138.19  $\pm$  1.18 km² (+81.7%). The development and evolution of lake-terminating glacier–proglacial lake systems are predominantly concentrated along the southern margin of HMA, including the Hindu Kush, Himalayas, Nyainqentanglha, and Gangdise Mountains.

This dataset offers a robust basis for examining spatially heterogeneous glacier responses to climate change, coupled glacier-lake evolution, glacier hydrological modeling, glacial lake outburst flood (GLOF) assessment, and water resource management. Nevertheless, further improvements in data quality

## https://doi.org/10.5194/essd-2025-596 Preprint. Discussion started: 20 October 2025 © Author(s) 2025. CC BY 4.0 License.

| 376 | remain necessary, particularly in quantifying glacier-lake contact line length, the degree of glacier-lake                      |
|-----|---------------------------------------------------------------------------------------------------------------------------------|
| 377 | contact (e.g., lake depth and subaqueous glacier front depth), and water temperature measurements.                              |
| 378 |                                                                                                                                 |
| 379 | Financial support. This work was funded by the National Key R&D Program of China (Grant No.                                     |
| 380 | 2024YFC3013400) and National Science Foundation of China (Grant No. 42361144874).                                               |
| 381 |                                                                                                                                 |
| 382 | $\textbf{Author contributions.} \ YL \ designed \ the \ study, \ developed \ the \ methodology, \ performed \ analysis, \ and$  |
| 383 | $wrote the \ manuscript. \ QL \ provided \ funding, \ support \ and \ supervision. \ XL, YY \ and \ JY \ produced \ data \ and$ |
| 384 | performed analysis. All other authors discussed and drafted the formulation of the specifications of the                        |
| 385 | glacial lake inventory in this study. All authors contributed to the final form of the paper.                                   |
| 386 | <b>Competing interests.</b> The authors declare that they have no conflict of interest.                                         |
| 387 |                                                                                                                                 |
| 388 | Code and data availability. Data described in this manuscript can be accessed at Zenodo under                                   |
| 389 | $https://doi.org/10.5281/zenodo.17369580 \ (Luo \ and \ Liu, \ 2025). \ The \ code \ for \ proglacial \ lake \ ident$           |
| 390 | $-ification\ can\ be\ accessed\ via\ https://code.earthengine.google.com/00573a1f3c8684d6f3e0722677f5$                          |
| 391 | 4b64.                                                                                                                           |
| 392 |                                                                                                                                 |
| 393 |                                                                                                                                 |

https://doi.org/10.5194/essd-2025-596 Preprint. Discussion started: 20 October 2025 © Author(s) 2025. CC BY 4.0 License.

Appendix A

Table A1 Glacier type statistics (number and area) across subregions

|                                |        |                |         |       |               |              |              | Glacier area (km²) | a (km²)       |             |               |               |
|--------------------------------|--------|----------------|---------|-------|---------------|--------------|--------------|--------------------|---------------|-------------|---------------|---------------|
| Region                         |        | Glacier number | in moet | 1     |               | 1990s        | s0           |                    |               | 2022s       | 2s            |               |
|                                | Type 1 | Type 2         | Type 3  | Total | Type 1        | Type 2       | Type 3       | Total              | Type 1        | Type 2      | Type 3        | Total         |
| Central Himalaya               | 94     | 138            | 84      | 316   | 589.97±2.81   | 264.93±1.64  | 219.8±1.17   | 1074.7±3.46        | 552.77±2.71   | 244.8±1.56  | 202.67±1.11   | 1000.25±3.32  |
| Westem Himalaya                | 99     | 106            | 28      | 200   | 198.31±2.11   | 176.41±1.67  | 42.54±0.47   | 417.26±2.74        | 192.5±2.08    | 167.24±1.64 | 40.28±0.45    | 400.02±2.69   |
| Eastern Himalaya               | 25     | 89             | 48      | 180   | 419.67±2.06   | 155.37±1.07  | 87.2±0.68    | 662.24±2.42        | 392.93±1.97   | 136.97±0.98 | 78.65±0.63    | 608.55±2.29   |
| Gangdise Mountains             | \$     | 77             | 50      | 161   | 35.19±0.32    | 45.65±0.37   | 33.34±0.33   | 114.17±0.59        | 31.79±0.31    | 40.11±0.34  | 29.53±0.31    | 101.44±0.55   |
| Hindu Kush                     | 61     | 75             | 12      | 148   | 26.49±0.31    | 45.82±0.41   | 8.71±0.17    | 81.02±0.54         | 23.58±0.29    | 41.34±0.39  | 7.83±0.16     | 72.74±0.51    |
| Nyainqentanglha                | 55     | 126            | 48      | 229   | 677.08±3.76   | 368.21±2.96  | 164.46±1.28  | 1209.75±4.95       | 650.54±3.73   | 350.51±2.91 | 154.73±1.22   | 1155.78±4.89  |
| Altun Shan/Eastern Kunlun Shan | 32     | 30             | 3       | 9     | 197.95±0.85   | 62.54±0.42   | 4.09±0.09    | 264.59±0.95        | 194.08±0.84   | 59.98±0.41  | 3.95±0.09     | 258.02±0.93   |
| Northem/Western Tien Shan      | 32     | 73             | 40      | 145   | 31.78±0.37    | 63.81±0.53   | 37.44±0.35   | 133.03±0.74        | 28.79±0.35    | 58.39±0.51  | 33.45±0.33    | 120.63±0.7    |
| Western Pamir                  | 30     | 59             | 17      | 901   | 93.74±0.95    | 99.99±0.84   | 16.58±0.29   | 210.3±1.09         | 89.95±0.93    | 94.04±0.8   | 15.33±0.28    | 199.33±1.26   |
| Central Tien Shan              | 25     | 49             | 17      | 16    | 733.67±10.9   | 53.48±0.64   | 29.86±0.6    | 817.01±10.94       | 729.83±10.9   | 49.1±0.61   | 27.78±0.57    | 806.71±10.93  |
| Qilian Shan                    | 21     | 7              | 10      | 38    | 119.22±0.92   | 11.38±0.19   | 59.35±0.54   | 189.95±1.08        | 116.6±0.9     | 10.31±0.18  | 57.66±0.52    | 184.57±1.05   |
| Eastern Tien Shan              | 18     | 46             | 118     | 82    | 33.23±0.54    | 39.06±0.38   | 29.91±0.54   | 102.2±0.85         | 30.49±0.52    | 35.32±0.36  | 27.46±0.52    | 93.27±0.82    |
| Karakoram                      | 18     | 18             | 9       | 42    | 66.51±0.78    | 102.9±0.99   | 128.82±2.25  | 298.24±2.58        | 65.1±0.76     | 100.62±0.97 | 128.1±2.25    | 293.82±2.56   |
| Tanggula Shan                  | 17     | 17             | 27      | 19    | 130.26±0.94   | 14.97±0.24   | 62.7±0.54    | 207.93±1.11        | 124.33±0.91   | 13.75±0.23  | 58.63±0.52    | 196.7±1.07    |
| Tibetan Interior Mountains     | 17     | 34             | 6       | 09    | 96.32±0.72    | 112.16±0.74  | 53.53±0.49   | 262.02±1.14        | 95.32±0.71    | 109.47±0.73 | 51.62±0.46    | 256.4±1.12    |
| Dzhungarsky Alatau             | 16     | 45             | 14      | 75    | 12.85±0.2     | 32.15±0.31   | 17.71±0.26   | 62.71±0.45         | 10.78±0.18    | 28.49±0.29  | 15.41±0.24    | 54.68±0.42    |
| Hengduan Shan                  | 16     | 99             | 27      | 108   | 10.75±0.18    | 49.63±0.5    | 27.82±0.31   | 88.2±0.61          | 8.03±0.16     | 42.71±0.47  | 22.99±0.27    | 73.72±0.57    |
| Pamir Alay                     | Ξ      | 29             | ∞       | 48    | 5.79±0.12     | 27.8±0.38    | 3.42±0.11    | 37±0.42            | 5.12±0.12     | 25.74±0.37  | $3.05\pm0.11$ | $33.91\pm0.4$ |
| Western Kunlun Shan            | 7      | 0              | 0       | 7     | 104.03±1.08   | 0            | 0            | 104.03±1.08        | 103.14±1.08   | 0           | 0             | 103.14±1.08   |
| Eastem Pamir                   | 2      | 5              | 1       | ∞     | 6.89±0.22     | 14.14±0.28   | 0.39±0.03    | 21.43±0.35         | 6.82±0.21     | 13.78±0.27  | 0.37±0.03     | 20.97±0.35    |
| Eastern Tibetan Mountains      | -      | 9              | -       | ∞     | 2.35±0.08     | 5.75±0.14    | 0.76±0.05    | 8.86±0.16          | 2.12±0.08     | 4.83±0.14   | 0.64±0.04     | 7.59±0.16     |
| Total                          | 299    | 1073           | 468     | 2208  | 3592.05±12.49 | 1746.17±4.43 | 1028.43±3.28 | 6366.64±13.65      | 3454.59±12.43 | 1627.49±4.3 | 960.13±3.18   | 6042.24±13.53 |

|                    | J      | Glacier number (1990s) | er (1990s) |       | ซี     | Glacier number (2022s) | er (2022s) |       |               | Glacier area (km²)(1990s) | (km²)(1990s) |               |               | Glacier area (km²)(2022s) | (km²)(2022s) |               |
|--------------------|--------|------------------------|------------|-------|--------|------------------------|------------|-------|---------------|---------------------------|--------------|---------------|---------------|---------------------------|--------------|---------------|
| Glacier size (km²) | Type 1 | Type 1 Type 2 Type 3   | Type 3     | Total | Type 1 | Type 2                 | Type 3     | Total | Type 1        | Type 2                    | Type 3       | Total         | Type 1        | Type 2                    | Type 3       | Total         |
| <0.05              | ∞      | 6                      | 4          | 21    | 20     | 33                     | 19         | 72    | 0.31±0.03     | 0.33±0.03                 | 0.15±0.02    | 0.78±0.04     | 0.72±0.04     | 1.05±0.05                 | 0.69±0.04    | 2.46±0.07     |
| 0.05-0.1           | 28     | 43                     | 26         | 76    | 37     | 72                     | 28         | 137   | 2.22±0.07     | 3.24±0.08                 | 1.86±0.06    | 7.32±0.12     | 2.72±0.08     | 5.36±0.11                 | 2.07±0.06    | 10.15±0.15    |
| 0.1-0.5            | 991    | 437                    | 136        | 739   | 168    | 442                    | 151        | 761   | 44.18±0.34    | 118.86±0.54               | 39.44±0.31   | 202.48±0.71   | 45.06±0.34    | 118.54±0.54               | 43.13±0.33   | 206.73±0.72   |
| 0.5-1              | 126    | 239                    | 101        | 466   | 118    | 205                    | 98         | 409   | 92.8±0.53     | 166.86±0.72               | 71.75±0.47   | 331.41±1.01   | 84.4±0.52     | 141.21±0.66               | 60.88±0.44   | 286.49±0.94   |
| 1-5                | 197    | 277                    | 154        | 829   | 188    | 256                    | 141        | 285   | 478.95±1.49   | 563.75±1.61               | 343.68±1.24  | 1386.38±2.52  | 459.59±1.47   | 515.28±1.55               | 318.5±1.18   | 1289.21±2.44  |
| 5-10               | 71     | 43                     | 30         | 4     | 29     | 14                     | 26         | 134   | 498.24±1.92   | 303.8±1.57                | 207.84±1.19  | 1009.87±2.75  | 455.09±1.82   | 287.6±1.51                | 180.33±1.09  | 927.97±2.56   |
| 10-50              | 63     | 23                     | 16         | 102   | 19     | 22                     | 16         | 66    | 1347.15±4.57  | 432.18±2.45               | 248.78±1.6   | 2028.11±5.43  | 1268.56±4.37  | 432.17±2.49               | 240.09±1.54  | 1931.43±5.26  |
| 50-100             | S      | 2                      | 0          | 7     | S      | 2                      | 0          | 7     | 343.42±2.96   | 157.16±2.78               | 0            | 500.58±4.06   | 332.14±2.97   | 155.79±2.76               | 0            | 487.93±4.05   |
| >100               | 3      | 0                      | -          | 4     | 3      | 0                      | -          | 4     | 784.78±10.96  | 0                         | 114.93±2.23  | 899.7±11.18   | 785.42±10.96  | 0                         | 114.44±2.22  | 899.86±11.18  |
| Total              | 299    | 1073                   | 468        | 2208  | 299    | 1073                   | 468        | 2208  | 3592.05±12.49 | 1746.17±4.43              | 1028.43±3.28 | 6366.64±13.65 | 3454.59±12.43 | 1627.49±4.3               | 960.13±3.18  | 6042.24±13.53 |

|                                |        | Lake number (1990s) | er (1990s) |       | T      | Lake number (2022s) | er (2022s) |       |                | Lake area | Lake area (km²) (1990s) |                 |               | Lake area (l  | Lake area (km²) (2022s) |               |
|--------------------------------|--------|---------------------|------------|-------|--------|---------------------|------------|-------|----------------|-----------|-------------------------|-----------------|---------------|---------------|-------------------------|---------------|
| Kegion                         | Type 1 | Type 2              | Type 3     | Total | Type 1 | Type 2              | Type 3     | Total | Type 1         | Type 2    | Type 3                  | Total           | Type 1        | Type 2        | Type 3                  | Total         |
| Central Himalaya               | 91     | 0                   | 98         | 177   | 91     | 149                 | 80         | 320   | 46.47±0.38     | 0         | 13.83±0.2               | 60.29±0.43      | 76.89±0.51    | 10.02±0.16    | 15.7±0.21               | 102.62±0.58   |
| Western Himalaya               | 92     | 0                   | 28         | 93    | 9      | 107                 | 27         | 199   | 4.17±0.11      | 0         | 1.2±0.05                | 5.38±0.12       | 9.02±0.15     | 4.13±0.09     | 1.51±0.06               | 14.66±0.19    |
| Eastern Himalaya               | 57     | 0                   | 90         | 107   | 99     | 75                  | 49         | 180   | 23.92±0.26     | 0         | 5.05±0.11               | 28.98±0.29      | 36.75±0.36    | 10.89±0.18    | 6.55±0.13               | 54.19±0.42    |
| Gangdise Mountains             | 99     | 0                   | 51         | 116   | 99     | 79                  | 51         | 195   | 3.72±0.1       | 0         | 3.19±0.08               | $6.91 \pm 0.13$ | 4.49±0.1      | 2.32±0.07     | 2.67±0.07               | 9.48±0.14     |
| Hindu Kush                     | 09     | 0                   | 12         | 72    | 19     | 75                  | 12         | 148   | 2.97±0.08      | 0         | $0.61\pm0.04$           | 3.58±0.09       | 4.83±0.11     | $1.81\pm0.06$ | 0.49±0.04               | 7.13±0.12     |
| Nyainqentanglha                | 52     | 0                   | 51         | 103   | 52     | 135                 | 49         | 236   | 12.53±0.18     | 0         | 5.78±0.13               | 18.31±0.22      | 28.57±0.3     | 8.16±0.14     | 7.52±0.14               | 44.25±0.36    |
| Altun Shan/Eastern Kunlun Shan | 33     | 0                   | 3          | 36    | 32     | 38                  | 3          | 73    | 4.83±0.14      | 0         | 0.17±0.02               | 5±0.14          | 5.68±0.14     | 1.06±0.05     | 0.11±0.02               | 6.85±0.15     |
| Northem/Western Tien Shan      | 32     | 0                   | 41         | 73    | 31     | 75                  | 39         | 145   | 1.15±0.05      | 0         | 1.44±0.05               | 2.6±0.07        | 2.3±0.07      | 1.97±0.06     | 1.51±0.06               | 5.77±0.11     |
| Western Pamir                  | 28     | 0                   | 19         | 47    | 28     | 09                  | 19         | 107   | 1.27±0.05      | 0         | $0.91\pm0.04$           | 2.17±0.07       | $2.81\pm0.08$ | 3.04±0.08     | 0.89±0.04               | 6.74±0.12     |
| Central Tien Shan              | 26     | 0                   | 17         | 43    | 25     | 49                  | 16         | 06    | $10.78\pm0.17$ | 0         | $0.71\pm0.04$           | 11.49±0.18      | 10.37±0.21    | 1.81±0.06     | $0.81\pm0.04$           | 13±0.22       |
| Qilian Shan                    | 20     | 0                   | Ξ          | 31    | 20     | ∞                   | Ξ          | 39    | 2.81±0.07      | 0         | $0.95\pm0.04$           | 3.76±0.09       | 4.24±0.1      | 0.26±0.02     | $0.88\pm0.04$           | 5.37±0.11     |
| Eastern Tien Shan              | 17     | 0                   | 20         | 37    | 17     | 47                  | 16         | 80    | 0.76±0.04      | 0         | $0.73\pm0.04$           | $1.49\pm0.05$   | 1.72±0.06     | 1.56±0.05     | $0.8\pm0.04$            | 4.08±0.09     |
| Karakoram                      | 19     | 0                   | S          | 24    | 19     | 17                  | 3          | 39    | 1.69±0.07      | 0         | $0.32\pm0.03$           | $2.02\pm0.07$   | 2.64±0.08     | 0.98±0.04     | 0.05±0.01               | 3.66±0.09     |
| Tanggula Shan                  | 17     | 0                   | 28         | 45    | 17     | 17                  | 26         | 09    | $3.1\pm0.08$   | 0         | 2.28±0.09               | 5.38±0.12       | 6.15±0.12     | 0.57±0.03     | 1.35±0.05               | 8.07±0.14     |
| Tibetan Interior Mountains     | 4      | 0                   | 6          | 23    | 14     | 35                  | 7          | 99    | 2.56±0.11      | 0         | $0.94\pm0.05$           | 3.5±0.12        | $3.58\pm0.14$ | 1.75±0.06     | $0.91\pm0.04$           | 6.23±0.16     |
| Dzhungarsky Alatau             | 14     | 0                   | 16         | 30    | 41     | 84                  | 15         | 77    | 0.37±0.03      | 0         | $1.17\pm0.05$           | $1.54\pm0.06$   | $1.03\pm0.05$ | 1.38±0.05     | 1.25±0.05               | 3.66±0.09     |
| Hengduan Shan                  | 15     | 0                   | 28         | 43    | 15     | 99                  | 27         | 108   | 0.71±0.04      | 0         | 1.63±0.06               | 2.35±0.07       | 1.26±0.06     | 1.84±0.06     | 1.91±0.07               | 5.01±0.11     |
| Pamir Alay                     | 12     | 0                   | ∞          | 20    | 12     | 32                  | ∞          | 52    | $0.5\pm0.03$   | 0         | $0.34\pm0.02$           | $0.84\pm0.02$   | 0.77±0.04     | 0.94±0.04     | 0.33±0.02               | 2.03±0.05     |
| Western Kunlun Shan            | 7      | 0                   | 0          | 7     | 7      | 0                   | 0          | 7     | 3.17±0.08      | 0         | 0                       | $3.17\pm0.08$   | 3.59±0.09     | 0             | 0                       | $3.59\pm0.09$ |
| Eastem Pamir                   | ю      | 0                   | -          | 4     | ю      | 5                   | -          | 6     | $0.22\pm0.03$  | 0         | $0.05\pm0.01$           | $0.26\pm0.03$   | 0.33±0.02     | $0.16\pm0.02$ | $0.04\pm0.01$           | $0.53\pm0.03$ |
| Eastem Tibetan Mountains       | -      | 0                   | -          | 7     | -      | 9                   | -          | œ     | $0.1\pm0.01$   | 0         | $0.04\pm0.01$           | $0.14\pm0.02$   | $0.19\pm0.02$ | $0.21\pm0.02$ | $0.02\pm0.01$           | $0.42\pm0.03$ |
| Total                          | 648    | 0                   | 485        | 1133  | 645    | 1123                | 460        | 2228  | 127.82±0.61    | 0         | 41.33±0.32              | 169.15±0.69     | 207.18±0.82   | 54.85±0.35    | 45.31±0.34              | 307.34±0.96   |

|                    |        |        |        | Lake number | ımper  |        |        |       |                  |        |            | Lake area (km²)                                                      | 2a (km²)    |            |            |             |
|--------------------|--------|--------|--------|-------------|--------|--------|--------|-------|------------------|--------|------------|----------------------------------------------------------------------|-------------|------------|------------|-------------|
| Glacier size (km²) |        | 1990s  | 0s     |             |        | 2022s  | 2s     |       |                  | 19     | 1990s      |                                                                      |             | 2022s      | 22s        |             |
|                    | Type 1 | Type 2 | Type 3 | Total       | Type 1 | Type 2 | Type 3 | Total | Type 1           | Type 2 | Type 3     | Total                                                                | Type 1      | Type 2     | Type 3     | Total       |
| 

| Table A.S. Artea changes of uniterent graciel types in each subregion (1770–1022) | 0            |                |            |                |        |              |        |       |
|-----------------------------------------------------------------------------------|--------------|----------------|------------|----------------|--------|--------------|--------|-------|
| Dordon                                                                            |              | Area loss(km²) | (km²)      |                |        | Area loss(%) |        |       |
| N cgross                                                                          | Type 1       | Type 2         | Type 3     | Total          | Type 1 | Type 2       | Type 3 | Total |
| Central Himalaya                                                                  | 37.2±3.91    | 20.13±2.26     | 17.13±1.62 | 74.46±4.8      | 6.31   | 7.6          | 7.79   | 6.93  |
| Western Himalaya                                                                  | 5.81±2.97    | 9.17±2.34      | 2.26±0.65  | 17.24±3.84     | 2.93   | 5.2          | 5.31   | 4.13  |
| Eastem Himalaya                                                                   | 26.74±2.85   | 18.41±1.45     | 8.55±0.93  | 53.7±3.33      | 6.37   | 11.85        | 8.6    | 8.11  |
| Gangdise Mountains                                                                | 3.39±0.44    | 5.54±0.5       | 3.8±0.46   | 12.73±0.81     | 9.65   | 12.14        | 11.41  | 11.16 |
| Hindu Kush                                                                        | 2.91±0.43    | 4.48±0.56      | 0.88±0.23  | 8.27±0.74      | 10.98  | 9.78         | 10.13  | 10.21 |
| Nyainqentanglha                                                                   | 26.54±5.29   | 17.7±4.15      | 9.72±1.77  | $53.96\pm6.95$ | 3.92   | 4.81         | 5.91   | 4.46  |
| Altun Shan/Eastern Kunlun Shan                                                    | 3.87±1.19    | 2.57±0.59      | 0.14±0.13  | 6.58±1.33      | 1.95   | 4.1          | 3.38   | 2.48  |
| Northern/Westem Tien Shan                                                         | 2.99±0.51    | 5.42±0.74      | 3.99±0.49  | 12.4±1.02      | 9.42   | 8.49         | 10.66  | 9.32  |
| Westem Pamir                                                                      | 3.79±1.33    | 5.95±1.16      | 1.24±0.4   | 10.98±1.81     | 4.04   | 5.95         | 7.51   | 5.22  |
| Central Tien Shan                                                                 | 3.84±15.42   | 4.38±0.88      | 2.08±0.83  | 10.3±15.47     | 0.52   | 8.19         | 26.9   | 1.26  |
| Qilian Shan                                                                       | 2.62±1.28    | 1.07±0.27      | 1.7±0.75   | 5.39±1.51      | 2.2    | 9.4          | 2.86   | 2.84  |
| Eastern Tien Shan                                                                 | 2.74±0.75    | 3.74±0.52      | 2.45±0.75  | 8.93±1.18      | 8.25   | 9.58         | 8.18   | 8.74  |
| Karakoram                                                                         | 1.41±1.09    | 2.29±1.39      | 0.72±3.18  | 4.42±3.64      | 2.12   | 2.22         | 0.56   | 1.48  |
| Tanggula Shan                                                                     | 5.94±1.31    | 1.22±0.33      | 4.07±0.75  | 11.23±1.55     | 4.56   | 8.18         | 6.49   | 5.4   |
| Tibetan Interior Mountains                                                        | 1.01±1.02    | 2.69±1.04      | 1.92±0.68  | 5.62±1.61      | 1.04   | 2.4          | 3.58   | 2.14  |
| Dzhungarsky Alatau                                                                | 2.07±0.26    | 3.66±0.43      | 2.3±0.36   | 8.03±0.62      | 16.1   | 11.38        | 12.99  | 12.8  |
| Hengduan Shan                                                                     | 2.73±0.24    | 6.92±0.69      | 4.83±0.41  | 14.48±0.84     | 25.34  | 13.95        | 17.37  | 16.42 |
| Pamir Alay                                                                        | 0.67±0.17    | 2.06±0.53      | 0.37±0.15  | $3.1 \pm 0.58$ | 11.55  | 7.4          | 10.75  | 8.36  |
| Western Kunlun Shan                                                               | 0.89±1.53    | 0              | 0          | 0.89±1.53      | 0.86   | 0            | 0      | 98.0  |
| Eastern Pamir                                                                     | 0.08±0.3     | 0.36±0.39      | 0.02±0.04  | 0.46±0.49      | 1.1    | 2.57         | 5.37   | 2.15  |
| Eastem Tibetan Mountains                                                          | 0.23±0.11    | 0.92±0.19      | 0.12±0.06  | 1.27±0.23      | 9.81   | 15.94        | 16.33  | 14.35 |
| Total                                                                             | 137.46±17.62 | 118.68±6.18    | 68.29±4.58 | 324.43±19.23   | 3.83   | 8.9          | 6.64   | 5.1   |

| Glacier size |        | Number change(count) | count) |       |               | Area change(km²) | ıge(km²)    |               |        | Area change(%) | nge(%) |        |
|--------------|--------|----------------------|--------|-------|---------------|------------------|-------------|---------------|--------|----------------|--------|--------|
| (km²)        | Type 1 | Type 2               | Type 3 | Total | Type 1        | Type 2           | Type 3      | Total         | Type 1 | Type 2         | Type 3 | Total  |
| <0.05        | 12     | 24                   | 15     | 51    | 0.41±0.05     | 0.73±0.05        | 0.54±0.04   | 1.68±0.08     | 133.95 | 224.58         | 367.1  | 215.87 |
| 0.05-0.1     | 6      | 29                   | 2      | 40    | 0.58±0.1      | 2.04±0.13        | 0.21±0.09   | 2.83±0.19     | 26.17  | 62.92          | 11.3   | 38.68  |
| 0.1-0.5      | 2      | 8                    | 15     | 22    | 1.67±0.48     | -1.1±0.77        | 3.68±0.46   | 4.25±1.02     | 3.78   | -0.93          | 9.33   | 2.1    |
| 0.5-1        | œ,     | -34                  | -15    | -57   | -8.41±0.74    | -25.65±0.97      | -10.87±0.64 | -44.93±1.38   | -9.06  | -15.37         | -15.15 | -13.56 |
| 1-5          | 6-     | -21                  | -13    | 43    | -19.48±2.11   | -52.52±2.21      | -25.17±1.71 | -97.17±3.5    | -4.07  | -9.32          | -7.32  | -7.01  |
| 5-10         | 4      | -2                   | 4      | -10   | -38.2±2.62    | -16.19±2.18      | -27.51±1.57 | -81.9±3.75    | -7.67  | -5.33          | -13.24 | -8.11  |
| 10-50        | -2     | 7                    | 0      | ٤-    | -63.39±6.38   | -24.61±3.4       | -8.69±2.22  | -96.69±7.56   | -4.71  | -5.69          | -3.49  | 4.77   |
| 50-100       | 0      | 0                    | 0      | 0     | -11.28±4.19   | -1.37±3.92       | 0           | -12.65±5.74   | -3.28  | -0.87          |        | -2.53  |
| >100         | 0      | 0                    | 0      | 0     | 0.64±15.5     | 0                | -0.48±3.15  | 0.16±15.82    | 0.08   |                | -0.42  | 0.02   |
| Total        | 0      | 0                    | 0      | 0     | -137.46±17.62 | -118.67±6.17     | -68.29±4.57 | -324.42±27.18 | -3.83  | 8.9-           | -6.64  | -5.1   |

Table A 6 Area and number changes of three glacier types across different size classes

Table A 7 Area changes of different glacial lake types in each subregion (1990–2022)

|                                |            | Area chan  | ge (km²)   |             | Area   | a change (% | <b>6</b> ) |
|--------------------------------|------------|------------|------------|-------------|--------|-------------|------------|
| Region                         | Type 1     | Type 2     | Type 3     | Total       | Type 1 | Type 3      | Total      |
| Central Himalaya               | 30.42±0.64 | 10.02±0.16 | 1.88±0.29  | 42.32±0.72  | 65.46  | 13.6        | 70.19      |
| Western Himalaya               | 4.84±0.19  | 4.13±0.09  | 0.31±0.07  | 9.28±0.22   | 115.95 | 25.8        | 172.62     |
| Eastern Himalaya               | 12.83±0.45 | 10.89±0.18 | 1.49±0.17  | 25.22±0.51  | 53.63  | 29.49       | 87.04      |
| Gangdise Mountains             | 0.77±0.14  | 2.32±0.07  | -0.52±0.11 | 2.57±0.19   | 20.7   | -16.32      | 37.21      |
| Hindu Kush                     | 1.86±0.13  | 1.81±0.06  | -0.11±0.05 | 3.55±0.15   | 62.57  | -18.1       | 99.15      |
| Nyainqentanglha                | 16.04±0.35 | 8.16±0.14  | 1.74±0.19  | 25.95±0.42  | 128.03 | 30.11       | 141.75     |
| Altun Shan/Eastern Kunlun Shan | 0.85±0.2   | 1.06±0.05  | -0.05±0.02 | 1.86±0.21   | 17.59  | -30.3       | 37.21      |
| Northern/Western Tien Shan     | 1.14±0.09  | 1.97±0.06  | 0.07±0.08  | 3.17±0.13   | 98.76  | 4.86        | 122.14     |
| Western Pamir                  | 1.54±0.09  | 3.04±0.08  | -0.02±0.06 | 4.57±0.14   | 121.66 | -2.2        | 210.24     |
| Central Tien Shan              | -0.41±0.27 | 1.81±0.06  | 0.11±0.06  | 1.51±0.28   | -3.8   | 15.59       | 13.14      |
| Qilian Shan                    | 1.43±0.12  | 0.26±0.02  | -0.07±0.06 | 1.61±0.14   | 50.93  | -7.34       | 42.8       |
| Eastern Tien Shan              | 0.96±0.08  | 1.56±0.05  | 0.07±0.06  | 2.59±0.11   | 126.23 | 9.62        | 174        |
| Karakoram                      | 0.94±0.11  | 0.98±0.04  | -0.28±0.03 | 1.64±0.12   | 55.52  | -86.41      | 81.3       |
| Tanggula Shan                  | 3.04±0.14  | 0.57±0.03  | -0.93±0.1  | 2.69±0.18   | 98.01  | -40.83      | 50         |
| Tibetan Interior Mountains     | 1.02±0.18  | 1.75±0.06  | -0.03±0.07 | 2.73±0.2    | 39.92  | -3.18       | 78.03      |
| Dzhungarsky Alatau             | 0.66±0.06  | 1.38±0.05  | 0.08±0.07  | 2.12±0.1    | 176.38 | 6.85        | 137.53     |
| Hengduan Shan                  | 0.55±0.07  | 1.84±0.06  | 0.28±0.09  | 2.66±0.13   | 76.97  | 17.13       | 113.23     |
| Pamir Alay                     | 0.27±0.05  | 0.94±0.04  | -0.02±0.03 | 1.19±0.07   | 54.26  | -5.82       | 141.49     |
| Western Kunlun Shan            | 0.42±0.12  | 0          | 0          | 0.42±0.12   | 13.26  |             | 13.26      |
| Eastern Pamir                  | 0.11±0.04  | 0.16±0.02  | -0.01±0.01 | 0.26±0.04   | 50.87  | -20.66      | 98.25      |
| Eastern Tibetan Mountains      | 0.08±0.02  | 0.21±0.02  | -0.01±0.01 | 0.28±0.04   | 77.14  | -27.51      | 199.92     |
| Total                          | 79.36±1.02 | 54.85±0.35 | 3.98±0.47  | 138.19±1.18 | 62.09  | 9.63        | 81.7       |

Table A 8 Area and number changes of proglacial lakes of three glacier types across different size classes

| CI                 |        | Number | change |       |            | Area cha   | inge(km²)  |             | Are    | ea change(% | <b>6</b> ) |
|--------------------|--------|--------|--------|-------|------------|------------|------------|-------------|--------|-------------|------------|
| Glacier size (km²) | Type 1 | Type 2 | Type 3 | Total | Type 1     | Type 2     | Type 3     | Total       | Type 1 | Type 3      | Total      |
| <0.05              | -141   | 887    | -44    | 702   | -1.91±0.16 | 20.58±0.19 | -1.36±0.16 | 17.31±0.3   | -25.07 | -18.14      | 114.49     |
| 0.05-0.1           | 49     | 155    | -1     | 203   | 3.44±0.19  | 10.81±0.15 | 0.11±0.17  | 14.36±0.29  | 52.27  | 1.71        | 110.23     |
| 0.1-0.5            | 47     | 72     | 21     | 140   | 14.61±0.44 | 13.19±0.17 | 5.47±0.32  | 33.27±0.57  | 45     | 30.77       | 66.22      |
| 0.5-1              | 17     | 6      | -3     | 20    | 11.15±0.4  | 4.43±0.14  | -2.76±0.21 | 12.82±0.48  | 55.66  | -32.72      | 45.04      |
| >1                 | 25     | 3      | 2      | 30    | 52.07±0.79 | 5.84±0.12  | 2.53±0.13  | 60.44±0.81  | 85.19  | 215.49      | 97.02      |
| Total              | -3     | 1123   | -25    | 1095  | 79.36±1.02 | 54.85±0.35 | 3.98±0.46  | 138.19±1.18 | 62.09  | 9.63        | 81.7       |

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
