# Peer review of "Rapidly Changing Lake-Terminating Glaciers in High"

_Earth System Science Data, 2025_

## Author Comment (AC1)

Dear Editor and Reviewer #1,

We sincerely thank you for your careful review of our manuscript and for providing valuable comments and suggestions. We have carefully considered all the feedback and revised the manuscript accordingly. Below we provide a detailed, point-by-point response to each comment.

Comment 1: *Steiner et al. have shared the global LGT. And, this work also produce the LGT in HMA in different time. I would suggest share those LGT. And please Emphasize in the abstract that the vector data are provided as shapefiles / GeoPackage with full attribute tables compliant with the RGI 7.0 vocabulary; this is not obvious to casual readers.*

Response: Thank you for pointing this out. Following your suggestion, we have provided further clarification in the abstract, explaining that the data is published on Zenodo in GeoPackage format, adhering to the internationally recognized RGI 7.0 data standard. This ensures that our data is easily accessible and usable by readers and other researchers.

Comment 2: *It is useful to share the GEE javascript/python code as a supplementary file or link to a public Github.*

Response: We have packaged the code and shared it as a supplementary file accompanying this manuscript.

Comment 3: *The minimum lake threshold was used $0.0036 km^2$ (4-pixel), which is justified and consistent with earlier regional work. However, I also noted that the uncertainty of proglacial lakes less than $0.01 km^2$ is larger than 40% in Figure 6. Consequently, I wonder what's the point of taking such a small threshold? I would suggest Keep the threshold at $0.01 km^2$*

Response: We sincerely thank the reviewer for raising this important point. We fully understand your concern regarding the minimum lake threshold, especially noting that Figure 6 shows the uncertainty of proglacial lakes smaller than $0.01 km^2$ exceeding 40%. This indeed indicates that small proglacial lakes have relatively large measurement errors. In fact, when we initially considered the threshold for lake area, we also discussed this issue extensively. Particularly around 1990, the availability of remote sensing data was limited, and the image resolution and quality were relatively low, which made the measurement of small lakes more uncertain.

Nevertheless, we ultimately chose to adopt a minimum threshold of $0.0036 km^2$ (equivalent to 4 pixels), based on several considerations:

Priority of research objectives: The primary goal of this study is to identify and screen lake-terminating glaciers. To achieve this comprehensively, it is necessary to maximize the detection of proglacial lakes. Even though very small lakes may have higher measurement uncertainty, their existence is crucial for determining whether a glacier terminates in a lake.

Actual existence of small lakes: In reality, some small glaciers do have proglacial lakes with areas below 0.01 km², and newly formed proglacial lakes are often small in their early stages. If we set the threshold at 0.01 km², these lakes would be excluded, which would affect the identification of lake-terminating glaciers.

Role of manual verification: To mitigate the impact of uncertainty in small lake measurements, we conducted manual verification. This allowed us to confirm the factual existence of these lakes and their contact with glaciers. Thus, even if the area measurement is uncertain, the existence of the lakes can be reliably established.

Acceptability of uncertainty: While the uncertainty of small lake areas is indeed larger, especially in the early period around 1990, this uncertainty does not affect the factual existence of the lakes. In other words, the uncertainty mainly influences quantitative precision but does not alter the qualitative identification of lakes. Considering that the primary objective of this study is the identification of lake-terminating glaciers, we regard this level of uncertainty as acceptable.

In summary, we selected a smaller threshold to ensure the completeness of the study and the achievement of its scientific objectives. Although small lakes may have larger measurement uncertainties, through manual verification and methodological adjustments, we can guarantee the factual existence of the lakes and thereby achieve comprehensive identification of lake-terminating glaciers. We believe this choice aligns with the purpose of the study and does not affect the main conclusions.

Comment 4: *A 2025 global LTG inventory (Steiner et al., 2025) identified 1912 LTGs in HMA (vs. 1740 in this study), with only 47.4% overlap. The study attributes this to temporal differences (2000$\pm$2 vs. 2022) but does not explore potential methodological discrepancies (e.g., Steiner et al.＇s use of ASTER data and expert cross-validation), limiting cross-dataset consistency.*

Response: Thank you for your valuable comment. In the revised manuscript, we have further discussed the sources of differences between our dataset and that of Steiner et al. (2025), including both temporal and methodological aspects. Specifically, we elaborated on the influence of different imagery periods (2000$\pm$2 vs. 2022) and the variations in data extraction methods and classification criteria. These clarifications

have been added to the discussion section to address the reviewer's concern regarding methodological discrepancies.

Comment 5: *Table format.* "Table 4:" *is different with Table 1,2,3. Figure format* "Figure1,2,3,4,","Figure 5:"*, Please check those format*

Response: Thank you for pointing out the formatting inconsistencies. We have carefully checked and revised the table and figure formats to ensure consistency throughout the manuscript.

Comment 6: *Discussion session is weak. The title of this Manuscript is rapidly changes of LTG, So, it is better to link LTG changes to climate drivers (such as how climate effect on regional variations in LTG loss) or others. Furthermore, LTG are link to hazards, so it is usefull to discuss/ass their risk. It would enhance the dataset's value for disaster risk management.*

Response: Thank you for your valuable suggestions. Based on your feedback, we have revised the discussion section to include a new subsection linking changes in lake-terminating glaciers (LTGs) to climate drivers. Additionally, we have expanded the discussion to explore the connection between LTG changes and associated hazards, particularly in the context of glacial lake outburst floods (GLOFs). Through this addition, our goal is to enhance the practical significance of our dataset and highlight its importance for disaster risk management.